# Using symmetry to elucidate the importance of stoichiometry in colloidal crystal assembly

Nathan A. Mahynski [1,3], Evan Pretti[2], Vincent K. Shen[1] & Jeetain Mittal [2,3]

We demonstrate a method based on symmetry to predict the structure of self-assembling, multicomponent colloidal mixtures. This method allows us to feasibly enumerate candidate structures from all symmetry groups and is many orders of magnitude more computationally efficient than combinatorial enumeration of these candidates. In turn, this permits us to compute ground-state phase diagrams for multicomponent systems. While tuning the interparticle potentials to produce potentially complex interactions represents the conventional route to designing exotic lattices, we use this scheme to demonstrate that simple potentials can also give rise to such structures which are thermodynamically stable at moderate to low temperatures. Furthermore, for a model two-dimensional colloidal system, we illustrate that lattices forming a complete set of 2-, 3-, 4-, and 6-fold rotational symmetries can be rationally designed from certain systems by tuning the mixture composition alone, demonstrating that stoichiometric control can be a tool as powerful as directly tuning the interparticle potentials themselves.

[1] Chemical Sciences Division, National Institute of Standards and Technology, Gaithersburg, MD 20899-8320, USA. [2] Department of Chemical and Biomolecular Engineering, Lehigh University, 111 Research Drive, Bethlehem, PA 18015-4791, USA. [3] These authors jointly supervised this work: Nathan A. Mahynski, Jeetain Mittal. Correspondence and requests for materials should be addressed to N.A.M. (email: nathan.mahynski@nist.gov) or to J.M. (email: jeetain@lehigh.edu)

In order to design colloidal systems which self-assemble into crystals of arbitrary complexity, the interparticle interactions between colloids are typically treated as degrees of freedom to be optimized[1–3]. In practice, this tuning can be achieved through various means, including particle charge, shape, and functionalization[4–10]. The breadth of this design space can be appealing, and previous research efforts have yielded a wide range of different structures via this route[11]. Unfortunately, interactions which may be theoretically optimal for creating a given target structure are often quite complex, involving multiple length scales and inflections at relatively large distances, making them difficult to realize experimentally. As an alternative approach to using a single colloidal component with a complex interaction potential, exotic lattices may be assembled using multiple components with a set of relatively simple pairwise potentials. A system in which each particle is unique is said to have addressable complexity[12–15]. In this particular case, it is necessary to select the mixture composition to provide precisely the correct number of components to assemble each structure.

For multicomponent mixtures in general, however, composition is a tunable thermodynamic parameter which is often overlooked in the context of self-assembly. Recent work has shown that stoichiometry can be exploited to make adjustments to the outcome of equilibrium self-assembly of binary mixtures of DNA-functionalized particles (DFPs)[16–18]. DFP systems provide a particularly useful framework to study these effects in multicomponent mixtures known as the multi-flavoring motif[18], which can be used to readily control the relative strengths of different pairwise interactions experimentally. However, the implications of stoichiometric control on stabilizing new phases, especially with an increasing number of components, have yet to be fully understood. It is especially unclear if changing stoichiometry alone can be used to direct assembly into different structures in the same way as changing pairwise interactions, since this requires knowledge of the phase diagram for each system of interest. In principle, if all possible structures which could appear on a phase diagram for a system were known a priori, their relative free energies could be calculated and the diagram constructed; yet, such a library of structures is difficult to obtain and its completeness is often unclear.

Indeed, predicting the stable crystal structure of a set of known constituents remains an outstanding challenge in condensed matter physics[19,20] and is a predominant barrier to the rational design of functional materials. Numerous mathematical and computational approaches have been developed to make this problem tractable, including random structure searching[21,22], optimization and Monte Carlo tools[23–29], evolutionary algorithms[30–35], and machine learning[36]. While powerful, the stochastic nature of these methods means that it is not possible to guarantee all relevant configurations and different symmetries have been considered. In certain cases where entropic considerations are significant, candidate structures can be found via direct enumeration schemes based on packing[17]. Complex network materials such as metal-organic frameworks, zeolites and other silicates, and carbon polymorphs often require more rigorous approaches and have been fruitfully enumerated through the use of topological methods[37,38] to identify crystalline nets and assess their chemical feasibility[20,39–43]. To our knowledge, however, such techniques have yet to be leveraged to explore multicomponent colloidal crystals.

To this end, we present a method based on symmetry to easily enumerate and refine candidate crystalline lattices with any number of components: one of the primary barriers to investigating the impact of stoichiometry on equilibrium self-assembly. We consider two-dimensional systems in this work to readily demonstrate the nature of our method; however, we emphasize that it is general and extensible to three-dimensional crystalline systems as well. Furthermore, there are many important technological applications for ordered two-dimensional materials including interfacial films, monolayers, porous mass separating agents, and structured substrates which require careful tuning of their crystalline structure[44–47]. Epitaxial growth and layer-by-layer assembly also require a detailed understanding of two-dimensional precursors to grow three-dimensional crystals[48–50]. By combining geometric information from symmetry groups with stoichiometric constraints, it is possible to more systematically search the energy landscapes of colloidal systems for candidate structures than with stochastic optimization methods alone. Ground state phase diagrams may thus be computed with relative ease and without a priori knowledge of possible configurations. Our results reveal how stoichiometry, without any changes to pairwise interactions, can be used to rationally control the symmetry of the resulting crystal lattices. We demonstrate how enthalpically dominated colloidal systems with only two or three components, interacting with simple isotropic potentials, can give rise to a wide range of structures, and how selection between close-packed and open structures can be performed by changing composition alone. Furthermore, the generality of our method suggests this tactic is applicable to a range of experimentally realizable colloidal systems and can provide useful routes to complex structures for the design of advanced materials.

## Results

**Combining symmetry and stoichiometry.** In order to understand how symmetry can be employed to aid in multicomponent crystal structure prediction, consider a primitive cell with periodic boundary conditions, as is typically employed for molecular simulations (cf. Fig. 1a). We may consider discretizing this cell into nodes upon which particles can be placed—although this is only an approximation to the continuous nature of configuration space, this assumption proves very convenient for generating candidate cells and will be relaxed later to make the method fully general. Generating all configurations for a multicomponent mixture on such a grid is effectively impossible for all but the smallest grids due to a combinatorial explosion of the number of possibilities as the size of the cell increases[32,51,52]. For instance, in our two-dimensional example, a discretization of the unit cell with area $A$ into equal subunits of size $\delta^2$ leads to $N_{config}$ total configurations:

$$N_{config} = \frac{(A/\delta^2)!}{(A/\delta^2 - N_{tot})! \prod_i N_i!}, \qquad (1)$$

where $N_i$ is the number of $i$-type species (such that $N_{tot} = \sum_i N_i$).

However, all two-dimensional crystals may be classified into one of 17 different planar symmetry groups, known as wallpaper groups[37,38,53]. In three dimensions, 230 space groups are required to describe all unique symmetries. Wallpaper groups describe the set of unique combinations of isometries (translation, rotation, and reflection) of the Euclidean plane containing two linearly independent translations. These operations act on a tile, or fundamental domain, to tessellate the plane. In addition to the p1 wallpaper group corresponding to the conventional periodic simulation cell discussed above, 16 additional groups exist with differing symmetries: a detailed summary of these groups and their fundamental domains is available in Supplementary Tables 1 and 2, and elsewhere[53]. Topology provides a compact representation of each group, known as an orbifold, which describes how to fold or wrap the fundamental domain to superimpose all equivalent nodes (cf. Supplementary Fig. 1)[38,54]. For p1, this is a torus; Fig. 1a demonstrates that wrapping a grid onto it brings nodes on separate edges and corners into contact, effectively

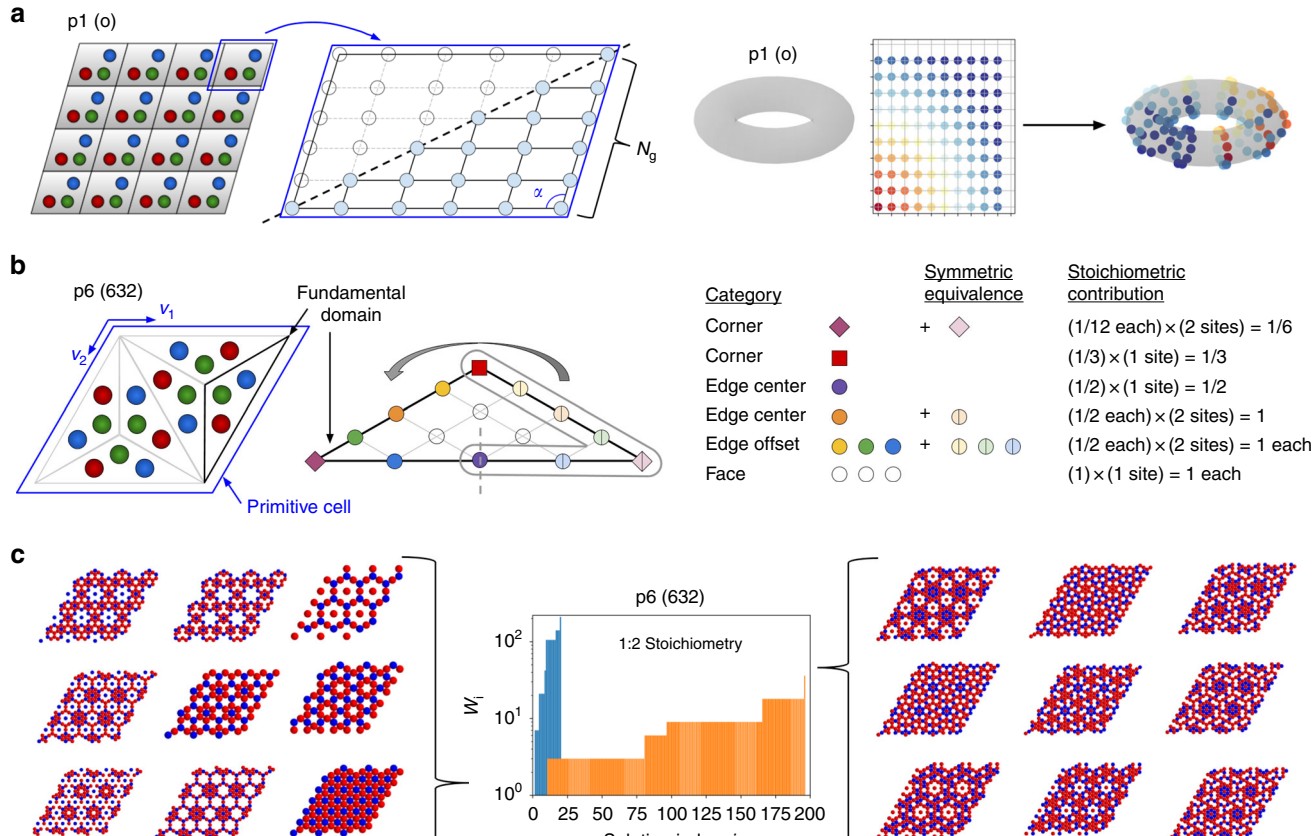

**Fig. 1** Summary of candidate enumeration strategy. **a** Discretization of a periodic p1 primitive cell into nodes. Symmetry introduces boundary conditions which constrain the placement of particles on the grid as if it were mapped onto the surface of a torus. This represents the orbifold of the p1 wallpaper group. **b** Fundamental domain and primitive cell for the p6 group; edge nodes are circled in gray on the right and will be wrapped on top of symmetrically equivalent nodes on the left. Here there are 6 node categories we could consider, which may be reduced to 4 if ones with the same stoichiometric contribution are combined. **c** Representative solutions to the CSP for the p6 lattice shown in **b** for a binary mixture with a 1:2 stoichiometric ratio of components. Images on the left are drawn from the solutions with the lowest 5% of realizations, and those on the right from the solution with the most. The blue histogram represents the case where node categories with the same stoichiometric contribution are combined, while the orange corresponds to when they are kept separate; the total number of solutions is the same for both

enforcing boundary conditions and constraining how particles may be positioned.

For each group there is a different set of connected fundamental domains that form the primitive cell, which contains the group's symmetries and may be used to cover the plane by translation operations alone. In groups other than p1, between 2 and 12 fundamental domains comprise the primitive cell[53]; thus, only a fraction of the primitive cell contains the independent configurational degrees of freedom in those groups, enabling a significant reduction of $A$ in Eq. (1). Consider for example p6, in Fig. 1b, in which the fundamental domain is triangular and has one sixth the area of the primitive cell. Furthermore, a large proportion of nodes are now found on the edges and corners, where symmetry-imposed boundary conditions cause some nodes to become equivalent to others. Our method leverages this, along with constraints due to stoichiometry, to achieve a significant computational advantage over the brute-force, combinatorial search method which uses only the p1 group. While colloids placed on face nodes are entirely contained within the domain, those at edge or corner nodes contribute only a fraction to its contents since they will be shared across multiple adjacent domains. Symmetrically equivalent boundary nodes may be collapsed to a single effective node with a net contribution equal to the sum of its equivalent nodes. Placing colloids over each group's fundamental domain may then be reduced to a constraint

satisfaction problem (CSP) in which the sum of the contributions from nodes where different colloid types are placed must satisfy a desired stoichiometric ratio (cf. Methods). The CSP is, in general, underspecified and admits many different solutions; each solution specifies how many of each type of colloid to place in different categories of nodes. For a $k$-component system with $n$ different node categories, the number of realizations of each different CSP solution, $W$, is

$$W = \prod_{j=1}^{n} \frac{C_j!}{(C_j - \sum_{i=1}^{k} m_{i,j})! \prod_{i=1}^{k} m_{i,j}!}, \qquad (2)$$

where $C_j$ refers to the number of nodes belonging to category $j$, and $m_{i,j}$ refers to the number of colloids of type $i$ assigned to nodes in that category. As a representative example, Fig. 1c shows the resulting solutions for a 1:2 stoichiometric ratio in a binary system for the p6 group.

Equation (2) is very similar to Eq. (1), and $W$ will also undergo a combinatorial explosion if $C_j$, the number of nodes in a category, $j$, is very large. However, relative to Eq. (1) this explosion is delayed by two factors. First, we have used symmetry to reduce the degrees of freedom by considering only the fundamental domain, which can be as little as one twelfth of the total primitive cell area. Second, we have reduced these degrees of freedom even further by using the symmetry of each group to

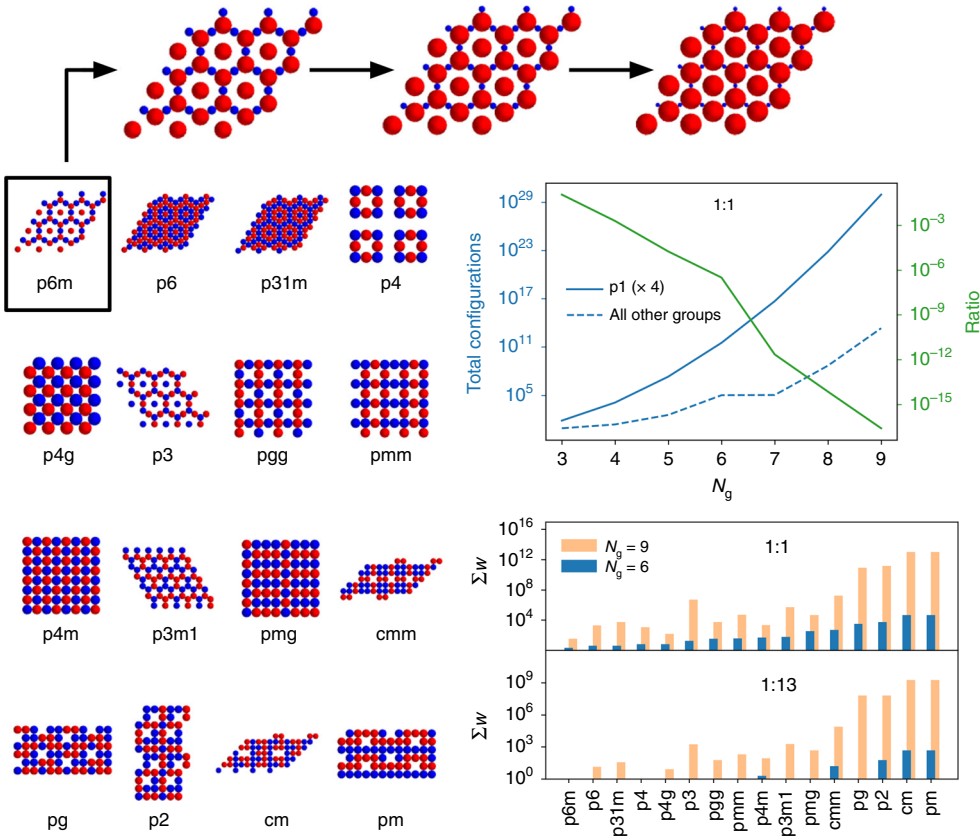

**Fig. 2** Enumeration of solutions to the constraint satisfaction problem (CSP). The total number of configurations, representing the sum of all realizations of all solutions to the CSP for all of the wallpaper groups besides p1, is given by the dashed blue line for a 1:1 binary mixture at various $N_g$. When not constrained by the symmetry of a given group, we set the sides of its fundamental domain equal to each other and $\alpha = \pi/2$. The solid blue line is the number of solutions for a total of 4 different p1 cells, each with different angles in their fundamental domain; the green line is the ratio between the two blue ones. Randomly chosen configurations for each group are also depicted which have been scaled to contact for equally sized colloids. A breakdown of the number of solutions each group contributes is also provided for representative $N_g$ values and stoichiometries. Above, the p6m group's solution has been scaled to contact assuming different diameters for the red colloids to illustrate how the same pattern can be used for differently sized colloids

remove edge nodes within a fundamental domain's lattice which are not independent. The second condition plays a significant role when the number of edge nodes relative to those on the face is large.

**Enumerating structures**. Combining symmetry and stoichiometry to cast the structure prediction problem as a CSP permits the tractable enumeration of crystalline configurations satisfying a given stoichiometric ratio up to moderately sized primitive cells. To see this, one may compute the number of nodes per edge of the fundamental domain for each group such that the nodal density approaches, but does not exceed, that of a chosen p1 reference cell. This reference cell is assumed to have $N_g$ nodes per edge and represents the case where no internal symmetry is present so that configurations are generated combinatorially without constraint. In our approach, an equally weighted average over all groups suggests that when $N_g \approx 8$ the total number of edge nodes will be equal to the number of face nodes (cf. Methods). For fundamental domains smaller than this, we expect that boundary symmetry for the groups will play a dominant role in determining valid configurations in the CSP. Taking the spatial discretization to also be on the order of the colloidal diameter, $\delta \sim \sigma$, the limiting p1 fundamental domain is on the order of $A \sim 8\sigma \times 8\sigma$. This is as large as boxes used to simulate many coarse-grained or colloidal fluids, implying that the upper bound for the primitive cell that can be feasibly generated with this method is reasonably large.

Examples of binary lattices generated by this scheme are presented in Fig. 2, along with a more concrete analysis of how it leads to a reduction in the number of possible configurations. Ternary examples can be found in the SI. In these cases, we have also allowed for the p1 reference cell parallelogram to be sheared to 4 different angles $\alpha \in [\pi/2, \pi/3, \pi/4, \pi/6]$ so that the resulting lattice is commensurate with other symmetries. Compared to p1, our approach to systematically enumerate non-trivial lattices over a similar area, i.e., size of primitive cell, for the other 16 wallpaper groups results in far fewer crystalline candidates that need to be considered. As anticipated, the total number of configurations does grow combinatorially at large $N_g$, which is dominated by lattices with a small number of fundamental domains per primitive cell (cf. SI); however, for $N_g \lesssim 8$, the total number of configurations is quite tractable.

For the binary system with a 1:1 stoichiometry shown in Fig. 2 there are less than $10^9$ configurations compared to an equivalent combinatorial search with the p1 cell, which results in $\mathcal{O}(10^{22})$ candidates when $N_g = 8$; this represents a reduction by over 13 orders of magnitude. A similar reduction occurs with ternary systems as well (cf. Supplementary Fig. 2). In both cases, the 1:1:(1) stoichiometry generates the most possible candidates; all other stoichiometries we investigated produced fewer solutions to the CSP, and thus $10^9$ configurations serves as a benchmark. A breakdown of these configurations into different groups is also shown, illustrating that for sufficiently small $N_g$ it is not possible to observe certain stoichiometries, which is expected from a

packing perspective. It is important to point out that the structures resulting from the 16 groups besides p1 are, in principle, a subset of the configurations resulting from the random search. This small subset composed of the other 16 groups contains additional symmetry beyond translation alone; this method simply enables those configurations to found directly rather than searching over all combinatorial realizations of where to place different colloids.

**Building phase diagrams.** To engineer the assembly of multi-component mixtures, their equilibrium phase behavior must be understood. We now illustrate how phase diagrams can be computed using this enumeration scheme. Specifically, we have applied this methodology to probe the self-assembly of mono-disperse colloidal monolayers formed from systems inspired by the multi-flavoring motif used in DFP assembly; this scheme enables all pairwise interactions in the system to become inde-pendent of one another, qualitatively ranging from being attrac-tive to repulsive. In the limit of strong binding, the ground state $(T^\star \rightarrow 0)$ serves as a reasonable approximation of the thermo-dynamically stable state[55]. Multi-flavored binary mixtures of colloids dominated by enthalpic interactions are known to exhibit a wide variety of morphologies, both experimentally and theoretically[18,55]; however, the full impact of stoichiometry on the thermodynamics of their self-assembly is not yet understood. Here we employ a simplified model (cf. Methods and Fig. 3a) to capture the tunability of the adhesiveness of arbitrary species pairs via a single parameter, $\lambda_{i,j}$, which ranges from $-1$ (repulsive) to $+1$ (attractive). This allows our model to maintain relevance beyond the specific case of DNA-mediated interactions; however, we emphasize that these kinds of interactions can be realized in various DFP systems, and that experimental results in such sys-tems are consistent with simulations employing potentials with pairwise tunable interactions[18]. Other, non-multi-flavored experimental DFP systems have also been successfully modeled with similar potential forms[56].

To predict the assembly of these mixtures, we first employed our scheme to enumerate a large number of the possible candidates within our framework. Although the grids constructed over the fundamental domains are consistent with each group's symmetry, they are artificial. Therefore, we subsequently relaxed these initially proposed candidates with a stochastic global optimization method known as basin hopping[25]. Note that lower symmetry structures which do not belong to any wallpaper group, such as quasicrystals, are not generally proposed in the initial candidate pool. A relaxation stage with basin hopping is therefore

important since it allows these lower symmetry structures to emerge from higher symmetry parent structures. Figure 3b illustrates an example where we have taken only the 25 candidates with the lowest energy from each group initially proposed (unrelaxed), and then performed this optimization procedure. The final, structurally unique lattices are plotted in the main panel, as only a few minima, including the ground state, tend to dominate the landscape and are found repeatedly. The ground state was often found multiple times by direct enumeration, which corresponds to the low-energy plateau in the inset. In fact, all stable periodic lattices reported in this work were found by direct enumeration, ultimately requiring no stochastic relaxation, demonstrating the robustness of this enumeration scheme.

For all sets of pairwise interactions, enumeration and optimization runs were performed for each canonical system corresponding to a fixed mole fraction. Phase diagrams were then computed by constructing the convex hull of (free) energy points in composition space (see schematic, Fig. 3c)[8,17,21]. States that lie on the hull are the thermodynamically stable states a system can attain, while all points above the hull represent metastable states. If a system's composition is prepared so it exactly matches one of the vertices on the hull, the associated structure will be produced. However, when the system's composition is intermediate between two vertices it will phase separate into the two corresponding structures, each with a different stoichiometry, as determined by the lever rule.

**Stoichiometric control.** Phase separation can, therefore, be har-nessed as a powerful mechanism for controlling self-assembly. A system with a fixed set of interparticle potentials that assembles into one structure out of a solution initially prepared at one composition, can give rise to a completely different structure when the solution is prepared with a different ratio of the same components. In this way, a single system can be designed so that simply by varying the solution mixing ratio of constituents, a number of structures with different stoichiometries can be pro-duced. Figure 4a shows a ground state phase diagram computed for a binary system. In Fig. 3b, we have illustrated how the square lattice is the lowest energy configuration for this system's set of pairwise interactions at a 1:1 stoichiometry; this forms the point on the convex hull at $x_1 = 0.5$ in Fig. 4a. However, other hon-eycomb phases intervene on the hull and enable the set of pair potentials to provide either square or honeycomb structures depending on the composition of the initial mixture.

We have validated our predictions using canonical molecular dynamics simulations, as shown in Fig. 4b. This demonstrates

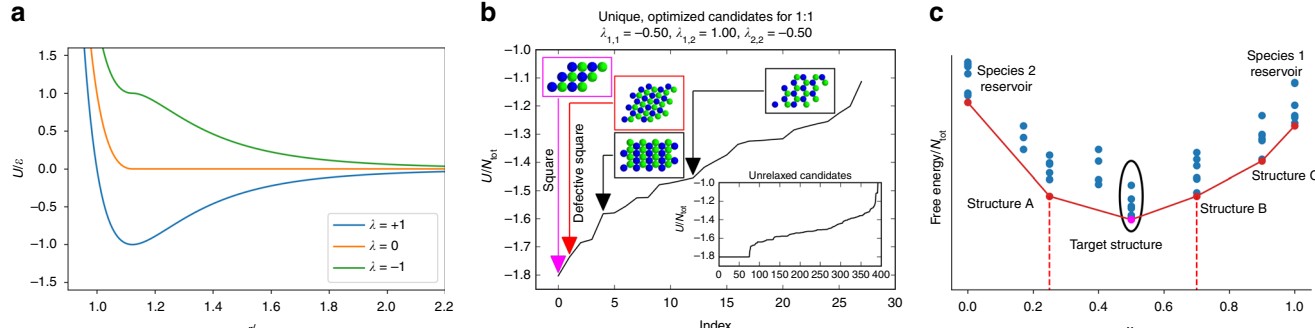

**Fig. 3** Construction of phase diagrams. **a** Potential energy function used in this work for representative values of the parameter $\lambda$. **b** Results of basin hopping (stochastic) optimization of an initial set of 25 candidates (inset) from each of the 16 groups considered, with a 1:1 stoichiometry. The main panel shows the specific energy of each structurally unique final candidate found, representing distinct local minima in energy. Several representative images are also shown. **c** Schematic of a representative convex hull (in red) showing thermodynamically stable structures on the hull as well as metastable structures (blue points). The ellipse encircles the points that result from the basin hopping stage as in **b**, with the magenta point corresponding to the ground state

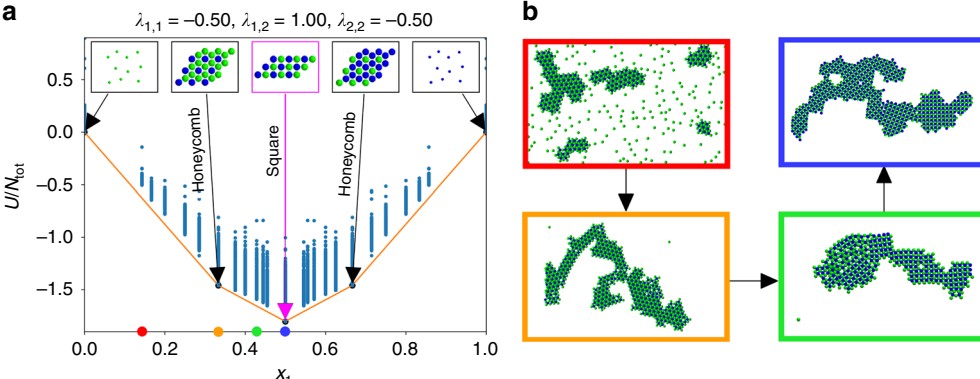

**Fig. 4** Constructing phase diagrams for a binary mixture. **a** Optimized results, as in Fig. 3b, from many stoichiometries can be combined to form a phase diagram. All points above the hull are metastable structures. **b** Representative snapshots from molecular dynamics simulations at $T^\star \approx 0.05$ for mole fractions of $x_1 \in [0.17, 0.33, 0.42, 0.50]$ depicted in red, orange, green, and blue, respectively, demonstrate the validity of the predictions in panel **a** and the ability of stoichiometry to affect assembly outcomes

that the phase diagram accurately represents the composition dependence of the system's behavior, and that this effect is realizable at finite temperatures and can be used to select structures under actual self-assembly conditions. The molecular dynamics results also show phase separation into the stable structures occurring when compositions between the vertices of the phase diagram are chosen. Note that in the red panel corresponding to $x_1 = 0.17$, a mixture of honeycomb crystals and a non-interacting vapor of particles have formed (expected since this species is self-repulsive, $\lambda_{1,1} = -0.50$). Similarly, in the green panel with $x_1 = 0.42$, a mixture of the honeycomb and square lattice structures are obtained. Clearly, for this system with a single set of pairwise interactions, changing the mixture stoichiometry from $x_1 = 0.33$ to $x_1 = 0.50$ allows for controlling assembly into these two different lattices which possess entirely different structural ordering and symmetry.

**Interactions vs. stoichiometry**. To understand the generality of this mechanism, we performed a broad survey of binary multi-flavored systems, computing phase diagrams at various $\lambda$, to elucidate how stoichiometry changes the relative stability of different lattices. We found a plethora of transitions that can be driven by stoichiometric effects alone, and overall, found that stoichiometric control can be as powerful as tuning the inter-particle interactions themselves. For a binary mixture there can be up to two coexisting phases in the ground state, and for each set of $\lambda = (\lambda_{1,1}, \lambda_{1,2}, \lambda_{2,2})$ values we considered, we report the most stable phase or phases as determined by the phase diagram constructed at those conditions. The key findings of this extensive set of calculations are summarized in Fig. 5.

In the ground state, the absolute values of the $\lambda_{i,j}$ do not matter, only the ratio of their values. In other words, a system where $\lambda = (0.25, 0.5, 0.2)$ will yield an identical structure to the case of $\lambda = (0.5, 1, 0.4)$. As a result, we can cast these $\lambda_{i,j}$ coordinates onto the surface of a unit sphere; in fact, since we are only concerned with the case where unlike species have a favorable interaction and will not simply phase separate into their pure component states ($\lambda_{1,2} \geq 0$), we need only consider one hemisphere. In Fig. 5, we report the structures found for three different representative stoichiometries. Unless explicitly shown, where the stoichiometry of the structures found is not equal to the composition of the solution, the remaining particles were found to coexist in an unstructured gas-like phase. In the parlance of Fig. 5, the fact that the color-coded structural changes occurring at a fixed $\lambda$ point between different mole fractions can be as dramatic as color-coded changes occurring at a fixed mole fraction as $\lambda$ is varied

illustrates that stoichiometric control (changing $x_1$) is as potent as engineering the potentials themselves (changing $\lambda$).

Transitions occurring in Fig. 5 are discussed at greater length in the Supplementary Discussion; however, the formation of the open honeycomb lattice is of special interest, as this 3-fold symmetric structure is an open, low-density lattice which is stabilized energetically, rather than entropically. In fact, although the pairwise interactions themselves follow a simple Lennard–Jones-like form, the ground state phase diagram contains numerous low-density lattices. When $x_1 = 0.66$ (2:1 stoichiometry), the lower left quadrant contains several cluster phases. In particular, where the open honeycomb structure was stable at $x_1 = 0.5$, now we find coexistence between extended rings, which follow a Kagome pattern (cf. Supplementary Fig. 8), and heptamer clusters. At $x_1 = 0.75$, these larger Kagome rings and heptamers give way to tetramer clusters. These predictions have been validated with molecular dynamics simulations as shown in Fig. 5. The native stoichiometry for this Kagome lattice is $x_1 = 0.6$, and once the mixture composition has been changed to this value, the system indeed forms only a single Kagome phase instead of coexisting with a second cluster phase. Self-assembly continues to occur well as density is increased up to its ideal value determined by that of the perfect lattice ($\rho^\star = 0.382$). Additionally, the square and various hexagonal phases have been realized in other simulations as well as experiments on multi-flavored DFP assembly[18], once again illustrating the consistency of this style of pairwise interactions with real physical systems, and the potential of this stoichiometric control scheme to be exploited for material design applications.

**Extension to ternary mixtures**. Among other transitions, Fig. 5 shows that tuning the stoichiometry alone can induce a ring opening event from a 3-fold open honeycomb lattice to an even lower density Kagome lattice in binary mixtures. To understand this further, and as a demonstration of our structure prediction approach for ternary systems, next we consider the impact of introducing a third component. We repeated our enumeration and optimization procedure for various ternary mixtures; as a representative result, here we restrict our discussion to the case where the third component is self-avoiding ($\lambda_{3,3} = -1$), yet interacts favorably with the second component ($\lambda_{3,2} = 1$) and essentially as a hard sphere with the first ($\lambda_{3,1} = 0$). Figure 6 summarizes the resulting phase diagram. We find that this third component can exert significant influence over the resulting morphology. While the ground-state phase diagram contains many different structures, a clear pattern emerges, which is

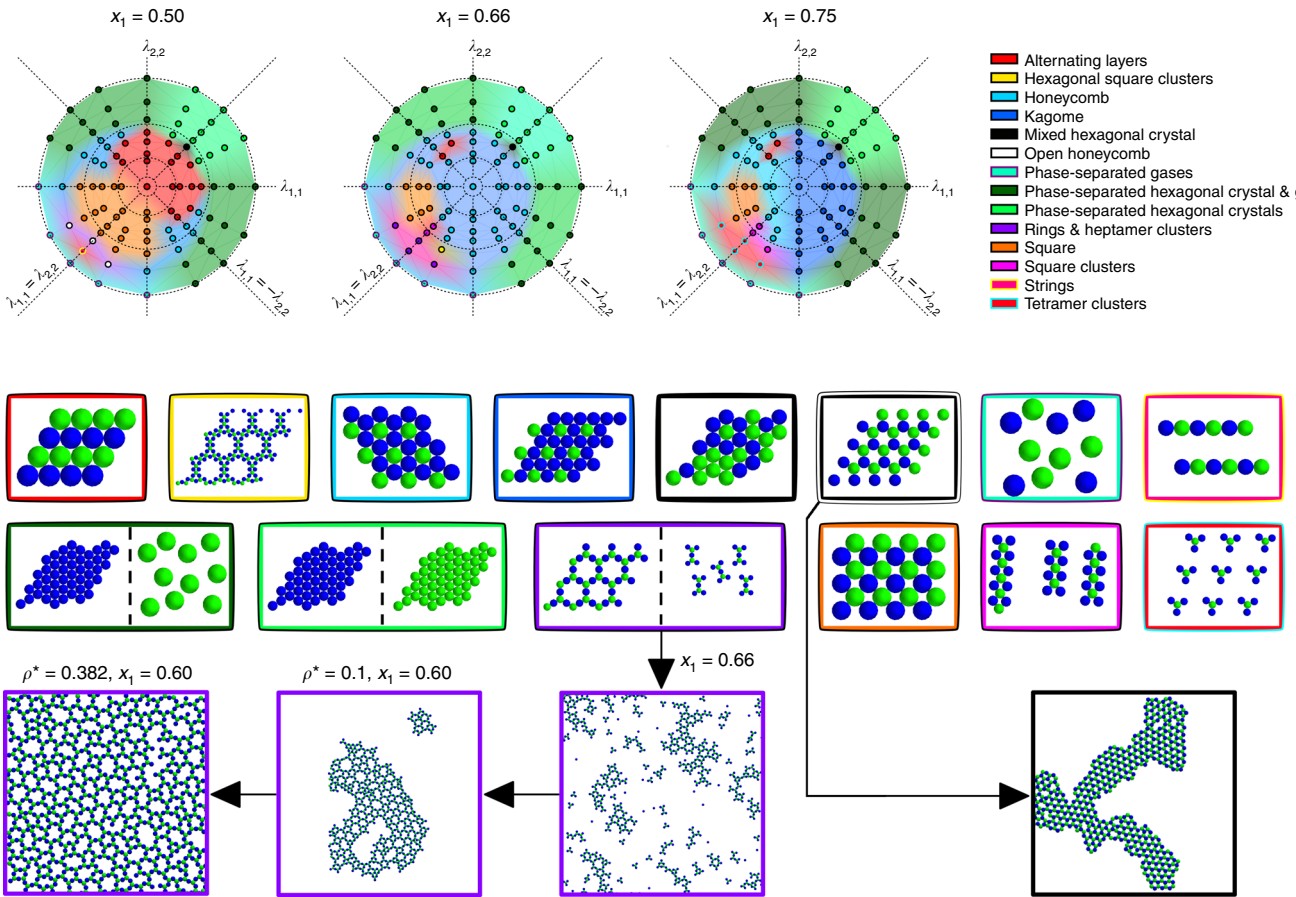

**Fig. 5** Thermodynamically stable structures for binary, multi-flavored colloidal mixtures. Various sets of $\boldsymbol{\lambda} = (\lambda_{1,1}, \lambda_{1,2}, \lambda_{2,2})$ were explored and mapped to a unit sphere. Only the projection of the hemisphere where $\lambda_{1,2} \geq 0$ is shown, where the outermost dashed circle denotes the equator ($\lambda_{1,2} = 0, \lambda_{1,1}^2 + \lambda_{2,2}^2 = 1$), and the central point denotes the zenith ($\lambda_{1,2} = 1, \lambda_{1,1} = \lambda_{2,2} = 0$). Structures were determined from the phase diagram computed at each $\boldsymbol{\lambda}$, and the results are indicated by a color-coded circle for each stoichiometry of species 1 (blue) and species 2 (green) considered. Depictions of the corresponding structures are outlined by the color of the point they correspond to; note that the choice of species coloring (blue vs. green) may be reversed for some regions of the diagram where one species is in excess of the other. The background coloring serves only for visualization purposes to guide the eye. Representative snapshots from molecular dynamics simulations of a few conditions are also depicted at the bottom

entirely controlled by the composition of the initial mixture. When species 3 is absent, the relative amounts of species 1 and 2 can be tuned to drive the system through transitions from gas-like phases (0-fold rotational symmetry), to clusters, to Kagome rings, to open honeycomb (3-fold rotational symmetry). Upon introducing species 3, depending on the composition of the parent solution, we may drive the system into 4-fold square lattices, 6-fold hexagonal ones, or even more extended ring structures (cf. Fig. 6a, b). The complete binary phase diagram is included for reference in Fig. 6c. These principal directions are highlighted by colored arrows and provide a basic compass for navigating the phase diagram (cf. Supplementary Discussion for more details).

We emphasize that this set of transformations, resulting in a complete range of rotational symmetries from gas-like (0-fold) up to hexagonal (6-fold) structures including low density rings and clusters, is brought about by changing the mixing ratio of the components alone. Furthermore, although temperature is expected to have a significant impact on the quantitative stability of different lattices, especially the cluster phases and rings, we achieved most of the predictions in molecular dynamics simulations at temperatures within an order of magnitude of the temperature at which we observed initial aggregation of the components. Thus, entropic contributions are not expected to change the qualitative conclusion that controlling stoichiometry in multicomponent mixtures can be a tool as powerful as

engineering the interparticle potentials for designing complex structures.

## Discussion

In summary, we have presented a method for investigating the stability of enthalpy-dominated multicomponent colloidal lattices and have used it to demonstrate that tuning the mixture composition can have as much impact as adjusting the interparticle potentials between the colloids themselves. Our approach is premised on recasting the structure prediction problem as a CSP in which symmetry and stoichiometry combine to form the constraints; the solutions to this problem, which may be enumerated and subsequently optimized with relative ease, are the candidate lattices to be considered. This method effectively generates a library of structures using only an upper bound for the size of the lattice's primitive cell and the desired stoichiometry. Such a library must otherwise be found by methods which are generally incomplete and prone to miss important candidates. In fact, every stable crystal structure reported in this work was found initially via enumeration, and subsequent optimization did not reveal additional stable candidates. This approach serves as an efficient way to explore all possible symmetry groups which helps ensure that the correct ground state is discovered.

It is important to highlight the general applicability of both the presented method and the results regarding stoichiometric

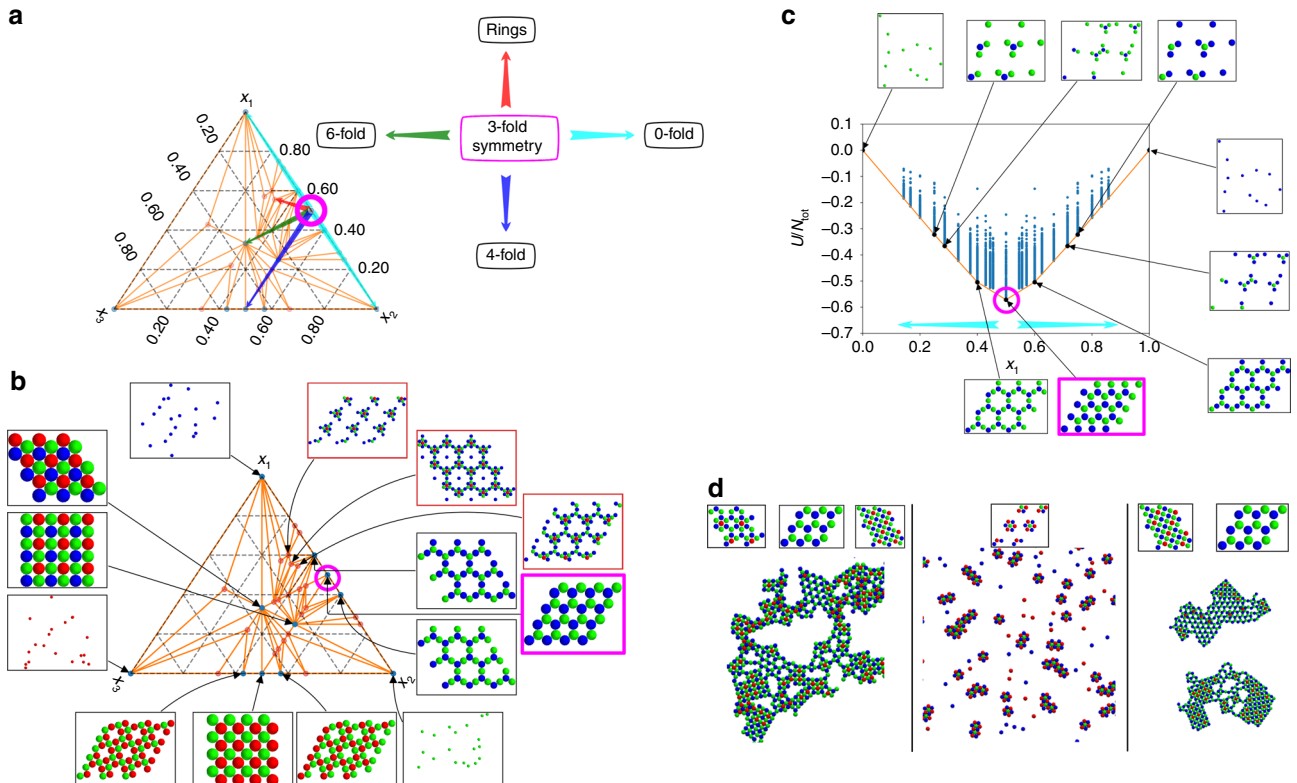

**Fig. 6** Ternary phase diagram for an example multi-flavored system. To a binary mixture where ($\lambda_{1,1} = -1.0$, $\lambda_{1,2} = 0.5$, $\lambda_{2,2} = -1.0$) a third component is added such that ($\lambda_{3,1} = 0$, $\lambda_{3,2} = 1$, $\lambda_{3,3} = -1$). **a** The triangular phase diagram is shown with arrows indicating directions along which the rotational symmetry of the most stable lattice is changed, starting from the open honeycomb (3-fold rotational symmetry) lattice. For reference, this structure is everywhere indicated in magenta. **b** Detailed diagram of structures that belong to the convex hull of (free) energy. Only landmark structures are depicted, in addition to certain very low density (ring-like) structures of interest which are outlined in red. More detail is available in the Supplementary Discussion. **c** The binary phase diagram for the initial mixture to which the third component was added. The cyan arrows correspond to those in **a**. **d** Molecular dynamics snapshots for various systems taken at $T^\star \approx 0.02$. From left to right, the stoichiometries, ($x_1$, $x_2$, $x_3$), are as follows: (0.40, 0.47, 0.13), (0.4285, 0.1430, 0.4285), (0.4165, 0.5000, 0.0835). The predicted structures from the phase diagram are shown above the snapshots

control. Although we have focused on presenting results from two-dimensional systems, the concepts presented here are extensible to higher dimensions as well. The interaction potentials considered here are very general, but experimental schemes for realization of such interactions in multicomponent systems exist using multi-flavored DFPs. These DFP systems are not limited to two dimensions, and simulations and experiments in both two[18,55] and three[57,58] dimensions have been performed on these systems to show the capacity of simple pairwise models to capture DFP assembly effects. They additionally demonstrate the feasibility of fine-tuning interactions in multicomponent mixtures as necessary to achieve self-assembly of particular structures. Finally, the results presented here have the potential to be particularly useful for physical realization of many superlattices including unique open structures, given that mixture stoichiometry is often easier to control than pairwise interactions, and has the potential to be just as powerful in terms of controlling structural ordering during self-assembly.

## Methods

**Creating regular grids on fundamental domains**. First, a regular grid, as depicted in Fig. 1a, is created over the surface of a group's fundamental domain. Nodes are placed along each edge with as close to the same spacing as possible such that there exist nodes at the termini of each edge. If this domain is triangular, the number of nodes along each edge must be identical so that interior nodes fall on the resulting parallelogram's diagonals. This happens regardless of the relative lengths of the sides. If this domain is a parallelogram, the number of nodes placed along adjacent edges may sometimes be different if the two sides have unequal lengths, as allowed

by symmetry constraints. This scheme covers different wallpaper groups differently, but in a consistent fashion which is commensurate with each group's unique symmetry.

Different groups have differently shaped fundamental domains, with the number of domains per primitive cell ranging from 1 to 12; therefore, we cannot simply place nodes at a fixed spacing along the edges of each group's fundamental domain and compute all possible resulting primitive cells as they would vary significantly in size. A more even-handed comparison can be made by working in reverse to compute the requisite grid spacings for each group's fundamental domain so that their primitive cells all cover a similar area. Although fundamental domains vary in shape, an approximate comparison may be made as follows.

As a reference, we consider a p1 primitive cell containing $N_g^2$ total nodes, and attempt to make the primitive cells of other groups have the same number of nodes. The number of nodes per edge of a group's fundamental domain may be estimated as

$$N_1 = \left\lfloor \sqrt{\frac{N_g^2}{r N_d \left(1 - \frac{1}{2} \times (N_s \bmod 2)\right)}} \right\rfloor, \quad (3)$$

where $N_1$ is the number of nodes along the shorter of the two edges which define the group's primitive cell, $r \geq 1$ is the ratio of the lengths of these edges, $N_d$ is the number of fundamental domains per primitive cell, and $N_s$ is the number of sides the fundamental domain has. The number of nodes on the longer edge is given by $N_2 = \lfloor r N_1 \rfloor$. A more detailed derivation is presented in the Supplementary Methods. The result is always a lattice that has no more than $N_g^2$ total nodes; consequently, $N_g$ should be viewed as a parameter that simply provides a way to compare the groups to each other by making their primitive cells congruent.

**The constraint satisfaction problem (CSP)**. For a system with $k$ total different colloid types, the number of times a colloid of type $i$ may be placed in a certain node category is $\mathbf{M}_i = (m_{i,1}, m_{i,2}, \ldots m_{i,n})$, which is a vector whose length is equal to the number of categories that exist on a given fundamental domain, $n$. If the

number of nodes in each category, $j$, is $C_j$, then $N_{\text{nodes}} = \sum_j^n C_j$, where $N_{\text{nodes}}$ is the total number of independent nodes on the fundamental domain. For the p6 group depicted in Fig. 1b, $N_{\text{nodes}} = 10$. The total number belonging to each category is bounded $0 \le \sum_i^k m_{i,j} \le C_j$, if we allow only one colloid per node.

Each node has a net fractional (stoichiometric) contribution, $\mathbf{F} = (f_1, f_2, \dots f_n)$, which is determined by symmetry and is independent of colloid type. For example, in Fig. 1b there are two distinct types of corners, one with $f_1 = 1/6$, and another with $f_2 = 1/3$. The total number of $i$-type colloids is $N_i = \mathbf{M}_i \cdot \mathbf{F}$; $\mathbf{F}$ is generally converted to whole numbers so that $N_i$ strictly contains integers. In principle, the categorization of nodes based on anything other than net fractional contribution is fictitious and those with the same value may be combined; in Fig. 1c the blue histogram shows this combined result, whereas the orange keeps the categories distinct. The total number of realizations, $\sum_i W_i$, is the same in both instances, and is on the order of $10^3$; however, keeping categories distinct can be advantageous. This tends to create more solutions, each with less individual realizations. When sorted by frequency, solutions with less combinatorial realizations tend to involve using fewer different categories of nodes, or nodes with special constraints, to solve the CSP. Consequently, solutions with less realizations (left side of Fig. 1c) tend to produce simpler structures which grow in apparent complexity as the number of solutions increases (right side).

We impose the constraints that at least one of each type of colloid must be placed somewhere, $N_i > 0 \; \forall i$, and require that the final ratio of $N_i$ values satisfies the desired stoichiometry, $\mathbf{S}_{\text{target}} = (1, N_2/N_1, N_3/N_1, \dots, N_k/N_1)$, where we have arbitrarily used $N_1$ to normalize. The value of $N_i$ is implicitly bounded above by the total number, and fractional contribution, of nodes available, though in principle this may also be constrained further. All $\bar{M} = (\mathbf{M}_1{}^{\mathrm{T}} \mathbf{M}_2{}^{\mathrm{T}} \dots \mathbf{M}_k{}^{\mathrm{T}})$, where it is understood that each $\mathbf{M}_i{}^{\mathrm{T}}$ forms a column in the $\bar{M}$ matrix, represent solutions that may be enumerated using a recursive backtracking algorithm. Each solution, $\bar{M}$, defines a prescription of how many of each type of colloid to place at each type of node. Some solutions will use only a small fraction of the available nodes, whereas others may employ them all.

All solutions to the CSP simply produce point patterns on lattices without any intrinsic length scale, and any lattice may be uniformly scaled without changing its symmetry. As a result, we choose to scale the resulting patterns to the contact point to produce the final candidate. For monodisperse, hard-sphere systems this is well defined. For other softer potentials one may use some characteristic length scale for the pairwise interactions; if there are multiple such length scales, e.g., multiple minima in the pairwise potential or if the system contains colloids of different diameters, multiple lattices can be generated from the same point pattern. The size-asymmetric case is illustrated in Fig. 2 with the p6m group. Also note that identical structures may sometimes be obtained from different groups as a given solution to the CSP may not use all of the edges or subtle features that distinguish groups from each other; e.g., consider the p6m and p3 for the binary case in Fig. 2. Each CSP solution does not violate any rules imposed by a group's symmetry constraints, but does not necessarily make use of them all (cf. Supplementary Methods for more details).

**Faces vs. edges of fundamental domains.** Consider a parallelogram with an equal number of nodes, $N_{\text{g}}$, along each edge. The number of nodes on the face, $N_{\text{f}} = (N_{\text{g}} - 2)^2$, exceeds the number of edge nodes, $N_{\text{e}} = 4(N_{\text{g}} - 1)$, when $N_{\text{g}} \ge 7$. For a triangular domain, $N_{\text{e}} = 3(N_{\text{g}} - 1)$ and $N_{\text{f}} = (N_{\text{g}} - 2)(N_{\text{g}} - 3)/2$, so that $N_{\text{g}} \ge 10$ represents this bound. In our systems there are 10 groups with parallelograms for fundamental domains, and 7 with triangular ones. Thus, an equally weighted average suggests that when $N_{\text{g}} \approx 8$, the number of edge nodes will be equal to the number of nodes on the face of the fundamental domain.

**Multi-flavored pairwise interactions.** The set of pairwise interactions used in this work are inspired by multi-flavored DFP systems[18,58,59]. With conventional DFPs, complementary strands of DNA are grafted on different particles inducing a favorable cross interaction due to DNA hybridization. However, when two colloids of the same type approach each other they simply repel each other as their grafts are identical. In multi-flavored systems, mixtures of different strands of complementary DNA are blended on the surfaces of different colloids decoupling the self- and cross-interactions in these mixtures. Controlling the surface composition of many different strands has the effect that one can independently tune the effective interactions between each pair of colloids. These enthalpy-dominated systems are typically assembled at relatively low density and ambient conditions[58,59], which corresponds to the limit where (osmotic) pressure effectively approaches zero. Furthermore, in the limit of strong binding, the ground state ($T^* \sim 0$) serves as a reasonable approximation of the system's thermodynamic state. In this case, the lowest energy lattice represents the most thermodynamically stable crystal as the Gibbs free energy is equal to potential energy in this limit.

To qualitatively model this behavior, all colloids interacted through a pair potential akin to Lennard–Jones which has been divided into its attractive and repulsive portions, then recombined according to some modulus, $\lambda_{i,j}$[55,60], which we refer to as the adhesiveness parameter:

$$U_{i,j}(r) = U_{i,j}^{\mathrm{r}}(r) + \lambda_{i,j} U_{i,j}^{\mathrm{a}}(r), \tag{4}$$

where

$$U_{i,j}^{\mathrm{r}}(r) = \begin{cases} 4\epsilon_{i,j}\left[\left(\frac{\sigma_{i,j}}{r}\right)^{12} - \left(\frac{\sigma_{i,j}}{r}\right)^6\right] + \epsilon_{i,j} & r \le 2^{1/6}\sigma_{i,j} \\ 0 & r > 2^{1/6}\sigma_{i,j}, \end{cases} \tag{5}$$

and

$$U_{i,j}^{\mathrm{a}}(r) = 4\epsilon_{i,j}\left[\left(\frac{\sigma_{i,j}}{r}\right)^{12} - \left(\frac{\sigma_{i,j}}{r}\right)^6\right] - U_{i,j}^{\mathrm{r}}(r). \tag{6}$$

The parameter, $\lambda_{i,j}$, effectively scales the energy from $U_{i,j}(2^{1/6}\sigma_{i,j}) = -\epsilon_{i,j}$ at $\lambda_{i,j} = 1$, to $U_{i,j}(2^{1/6}\sigma_{i,j}) = +\epsilon_{i,j}$ at $\lambda_{i,j} = -1$ (cf. Fig. 3a)[55,60]. Each pair of interactions has its own $\lambda_{i,j}$ value which may be tuned independently, mimicking the multi-flavoring motif of DFPs[58,59]. As a result, the characteristic contact point for this model is taken to be $2^{1/6}\sigma_{i,j}$. All colloids were given equal diameters, $\sigma_{i,j} = \sigma$, and energy scales, $\epsilon_{i,j} = \epsilon$. Only the value of $\lambda_{i,j}$ was varied to control the relative degree to which pairs of colloids attracted or repelled each other. Thus, all units reported herein are given in terms of $\epsilon$ and $\sigma$. All interactions were cut off at $r_{\text{c}} = 3\sigma$.

**Sampling wallpaper ensembles.** As described previously, a grid is generated for each wallpaper group in question, over which the CSP defined by the desired stoichiometric ratio of components in the final structure is solved recursively to enumerate all solutions. For wallpaper groups where the ratio between the lengths of the fundamental domain's sides, $r$, is not constrained by symmetry, we sampled $r \in [1, \sqrt{2}, \sqrt{3}, 2]$. In addition, for groups where the angle $\alpha$ between two adjacent edges of the fundamental domain is not constrained by symmetry, we generated all realizations of $\alpha \in [\pi/2, \pi/3, \pi/4, \pi/6]$. Each solution to the CSP yields a prescription to place a certain number of each colloid type on different types of nodes in the fundamental domain; the total number of realizations of each prescription is given by combinatorially choosing the number of colloids to be placed at each designated location. In all cases, we discarded p1 as the trivial method of prediction which quickly undergoes a combinatorial explosion for even a small grid. For all stoichiometries of interest in the binary mixture, we considered three cases: $N_{\text{g}} = 6$, $N_{\text{g}} = 8$, and a variable $N_{\text{g}}$. When $N_{\text{g}} = 6$ we exhaustively enumerated all primitive cells. These were scaled so that the minimum distance between colloids was $2^{1/6}\sigma$, then ranked based on energy; the lowest energy candidates from each group were subsequently refined with basin hopping. We did not repeat fully exhaustive sampling for $N_{\text{g}} = 8$, instead taking only 50,000 realizations of primitive cells from each group. Subsequent ranking and optimization yielded identical results. As a final check we also allowed $N_{\text{g}}$ to be variable, increasing to the point where each group yielded at least $10^5$ solutions to the CSP; from these ranked candidates we drew the best 100 structures from each group and optimized them. Again, the final results were the same.

**Basin hopping.** Basin hopping is a stochastic optimization approach well-suited to locating the global minimum of systems with hundreds of degrees of freedom and local minima separated by large barriers[25,61,62]. Atomistic, molecular, and colloidal systems often fall in this category and we adopted this approach here. The primitive cell is constructed from the fundamental domain according to the prescription provided by the CSP, which is the cell that is optimized. Basin hopping follows an iterative procedure where each cycle is composed of a perturbation followed by a deterministic relaxation to generate a new candidate, which is accepted as the new state of the system stochastically; here we used a Metropolis acceptance criterion:

$$p_{\text{acc}} = \min\left[1, \exp\left(-\frac{u_{\text{final}} - u_{\text{initial}}}{\hat{T}}\right)\right], \tag{7}$$

where $u$ is the potential energy per particle in each state and $\hat{T}$ is a parameter which controls the rate of acceptance. This was usually set to $\hat{T} = 0.50$ but is not related to the system's actual temperature, which in the ground state, is zero. Up to $10^4$ iterations were used to optimize each structure. We used the L-BFGS-B algorithm[63] to relax the initial candidate structure before performing the basin hopping, which also employed this algorithm. The total potential energy is a function of the coordinates, $\mathbf{r}$, of each of the $m$ colloids present and the primitive cell's vectors, $U = f(\mathbf{r}_1, \mathbf{r}_2, \dots, \mathbf{r}_m, \mathbf{L}_1, \mathbf{L}_2) = f(\psi)$; all variables in $\psi$ were optimized simultaneously. For precision, the candidate structure with the lowest energy resulting from basin hopping was further minimized with the Nelder–Mead simplex method[64] to achieve the final result.

Perturbation moves consisted of displacing a set of randomly chosen colloids, exchanging the locations of a randomly chosen set of pairs of colloids, perturbing the cell's vectors, shearing the cell, uniformly scaling the cell, and displacing local clusters of colloids as determined by a $k$-means algorithm[65]. These typically occurred with a 4:2:1:1:1:1 ratio. After each perturbation the cell's vectors were iteratively checked to find a more orthorhombic unit cell, if possible, to reduce the number of nearest neighbor images needed to compute the energy of the cell. This

is done by computing the distortion factor, $\mathcal{C}$[29,30]:

$$\mathcal{C}(\mathbf{L}_1, \mathbf{L}_2) = \frac{1}{4}(\| \mathbf{L}_1 \| + \| \mathbf{L}_2 \|)\frac{\mathcal{P}}{2A}, \qquad (8)$$

where $\mathcal{P}$ is the perimeter of the cell, and $A$ is its area. For a given primitive cell, new vectors are subsequently proposed: $(\mathbf{L}_1 \pm \mathbf{L}_2, \mathbf{L}_2)$, $(\mathbf{L}_1, \mathbf{L}_2 \pm \mathbf{L}_1)$. $\mathcal{C}$ is recomputed for each of these candidates, and the lattice with the lowest $\mathcal{C}$ is taken. This process is repeated until either $\mathcal{C}$ is not reduced by an iteration, or falls below a threshold of $\mathcal{C} \leq 1.5$. No more than 10 iterations are performed. A square cell has $\mathcal{C} = 1$. Importantly, symmetry was not constrained during optimization which allows an initially proposed structure to transform from one group into another and allows lower symmetry structures to emerge from higher symmetry parents.

**Structural similarity**. Various methods exist for determining the structural similarity of lattice configurations[66–72]. Our algorithm does not depend on differentiating lattices; however, we often removed similar structures from the final optimized set of non-ground-state structures to reduce the number to be examined a posteriori. This screening may also be performed as an intermediate stage to remove proposed candidates to be sent to basin hopping that may be considered too similar and therefore redundant. We employed radial distribution functions to determine this similarity, as this information is readily available on-the-fly following pairwise energy calculation. For two configurations, denoted $\alpha$ and $\beta$, we consider the cosine similarity of each colloid type to produce a vector, $\mathbf{S} = (S_{1,1}, S_{1,2}, \ldots, S_{n,n})$, such that

$$S_{i,j} = \frac{\mathbf{g}_{i,j}^\alpha(r) \cdot \mathbf{g}_{i,j}^\beta(r)}{\left\|\mathbf{g}_{i,j}^\alpha(r)\right\|\left\|\mathbf{g}_{i,j}^\beta(r)\right\|}, \qquad (9)$$

where $g_{i,j}(r)$ denotes the radial distribution function for the $(i, j)$ pair. We consider two configurations to be only as similar as their least similar pair; thus, $S = \min[\mathbf{S}]$ and we consider two configurations to represent different structures if $S < 0.90$. A more restrictive threshold of $S < 0.99$ did not change our final results. The radial distribution functions were computed out to a cutoff of $r_{\text{cut}} = 3\sigma$ with bins of width $\delta r = 0.2\sigma$.

**Phase diagrams**. Convex hulls of total energy per particle vs. mole fraction(s) were constructed using the QuickHull algorithm[73], as implemented in SciPy[62]. All points along the hulls reported were checked for energetic degeneracy, that is, unique structures that had energies per particle within $\delta(U/N_{\text{tot}}) \leq 10^{-6}$; no degeneracies were found for any of the conditions reported here. For ternary mixtures, the three-dimensional hull of $U/N_{\text{tot}}$ vs. $x_1$ and $x_2$. is projected onto the $(x_1 - x_2)$-plane; the faces of the three-dimensional hull indicate which phases are in coexistence and are depicted by the orange lines in Fig. 6, where the vertices correspond to the structures on the hull. All unique, integer stoichiometries $\xi_1{:}\xi_2$ up to $\xi_i \leq 6$ were considered for binary mixtures, and similar bounds were used for ternary mixtures (cf. Supplementary Discussion).

**Molecular dynamics**. Canonical (NVT) molecular dynamics simulations were performed in LAMMPS[74] using a Langevin thermostat with a time constant $\tau = \sigma m^{-1/2}\epsilon^{-1/2}$. Simulations were run with at least $10^3$ particles for at least $10^8$ timesteps, with each step $\Delta t = 10^{-2}\tau$. Numbers of components were rounded from the desired stoichiometric ratios to nearest integers. Temperatures $T^* = \epsilon^{-1}k_{\text{B}}T$ and number densities $\rho^* = \rho\sigma^2$ were set as desired. Here, $k_{\text{B}}$ is the Boltzmann constant and $T$ is the absolute temperature. Initial configurations were generated by random placement of particles, followed by energy minimization and equilibration at $T^* = 1$ for $10^5$ timesteps. The potential described previously, with a cutoff $r_{\text{c}} = 3\sigma$, was used.

## Data availability
Data is available on reasonable request. Direct requests to N.A.M. and J.M.

## Code availability
Code is available on reasonable request. Direct requests to N.A.M. and J.M.

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

## Acknowledgements

This work was supported by the U.S. Department of Energy, Office of Basic Energy Science, Division of Material Sciences and Engineering under Award (DE-SC0013979). E.P. acknowledges support from the National Institute of Standards and Technology Summer Undergraduate Research Fellowship (NIST SURF) program with Grant No. 70NANB16H. This research used resources of the National Energy Research Scientific Computing Center, a DOE Office of Science User Facility supported under Contract No. DE-AC02-05CH11231. Use of the high-performance computing capabilities of the Extreme Science and Engineering Discovery Environment (XSEDE), which is supported by the National Science Foundation, Project No. TG-MCB120014, is also gratefully acknowledged. Contribution of the National Institute of Standards and Technology, not subject to US Copyright.

## Author contributions

N.A.M., E.P., V.K.S., and J.M. designed the research. N.A.M. and E.P. performed the simulations and analyzed the data. All authors contributed to the writing of the manuscript.

## Additional information

**Competing interests:** The authors declare no competing interests.

