## [Peer Review File · Nature Communications]

Reviewers' comments:

Reviewer #1 (Remarks to the Author):

I have read the manuscript "Using symmetry to elucidate the importance of stoichiometry in colloidal crystal assembly" by Mahynski et al. with a great interest. The authors propose that the use of multicomponent systems with very simplified isotropic interparticle interactions can generate a number of complex 2D architectures. I, however, doubt how realistic are the assumptions used in the simulations.

Excluded volume entropy seems to be completely neglected in the simulations. Entropy is the only driving force in crystallisation of hard-wall colloids and it is an essential driving force for the majority of colloidal systems exhibiting spontaneous crystallisation. Assuming purely enthalpy-driven situation severely limits the applicability of the model to experimentally-realizable colloidal systems.

Do I understand it correct that Equation (5) is applicable also for $r < \sigma$? No hard wall interaction assumed?

The present simulation results are limited to 2D. Is there any experimentally-realizable analogue that can test the current predictions?

As I am not performing simulation myself, it is difficult for me to judge the advantage of the computational approach such as the constraint satisfaction problem. The observations are interesting. However, it is not clear to what extent they could be experimentally realised as the interparticle interactions seem to be unrealistic for colloidal particles.

In addition, spontaneous formation of complex structures by colloids with isotropic interactions was earlier demonstrated even in a single-component systems, using a combination of short-range attraction and long-range repulsion. I missed mentioning of these results in the introduction.

To summarize, I vote against publication of this MS in Nature Communications and suggest to send it to a journal specialised in computational techniques. At the same time, I encourage the authors to consider expanding their approach towards more realistic models, that could prove relevant for experimentally-realizable systems.

Andrei Petukhov

Reviewer #2 (Remarks to the Author):

This paper reports a new method for the search of two-dimensional colloidal crystals based on symmetry considerations. This is of course very interesting for the community of researchers working in crystal structure prediction. I leave to the editor to judge whether this paper can be of interest to a broader community. In any case, the paper is of high quality, so that I recommend publication after minor revision as specified below.

- Since this method is based on direct enumeration, I would like the authors to comment a little more deeply about its application to three-dimensional crystals, in which a third dimension may increase the number of possibilities in such a way that the method may become not applicable.

- The claim that short-range interactions mean small unit cells is not really granted. I do not see what long-range interactions stabilize quasi-crystals for example.

Reviewer #3 (Remarks to the Author):

Report for "Using symmetry to elucidate the importance of stoichiometry in colloidal crystal assembly" by Mahynski et al.

The authors address the problem of elucidating possible crystal structures formed by

multicomponent colloidal systems, in which the interactions can be tuned, and thus in principle the range of crystal structures accessible is very large. The authors claim to massively simplify this problem by considering the symmetries of all wallpaper groups in 2d, thus building a comprehensive set of structures. Notably, the use of symmetry as employed by the authors allows a spectacular drop in the number of structures that need to be considered. So if the results really are as impressive as the data seem to indicate, given the broad interest of colloidal (and nanoparticle) self-assembly, I think there is potential for this paper in a wide-ranging journal like Nature Communications.

However, in its present form, the manuscript is not easy to understand and much remains to be done in order to render it in a suitably palatable form. To give just one example, I had to go carefully through the SI to obtain absolutely essential results to understand the main message of the manuscript. The authors have been over-ambitious in terms of the amount of material they have chosen to attempt to present. They absolutely must take a lot of material out (like 50% say) if they are to have any hope of reaching the kind of general audience that Nature Communications aims at.

(1) In their literature review the authors have missed major contributors to this problem - Peter Harrowell in computer simulation and theory - and Steve Granick in experiment. Granick's kagome lattices [Nature 469 381 (2011)] are among rare (i.e. unique as far as I know) as non-trivial 2d colloidal self-assembly.

Meanwhile Harrowell has a series of papers with Toby Hudson, which tackled the same problem in 3d, starting from the set of all known crystals that form in atomic systems, focusing on inorganic materials, such as alloys. While of course such an approach is limited in terms of the dataset, one may observe that nature has had some very considerable time to sample the phase space. Examples include J. Phys. Chem. B 112, 10773 (2008).

Harrowell has explored other ways to reduce the configurational space than the approach of the authors - with Pierre Ronceray, he has used a lattice model to elucidate entire sets of structures in 3d: arXiv:1606.02579

Harrowell's methods are important, not least because they pertain to 3d. I accept that 2d is much easier for the authors to work with than 3d, but they cannot be as dismissive of 3d as they are in the current version of the manuscript.

Surely for sufficiently small primitive cells, 3d is tractable. As for the relevance of the size of primitive cell for "sufficient completeness", please see below - the author's claim that the size of the cells they consider is sufficient is far from justified. The authors should at least demonstrate that their approach is pertinent to 3d, by taking some small primitive cells.

(2) A major concern with the manuscript is the size of the primitive cell. The authors claim this should be somehow connected to the interaction range. This cannot be correct. See for example PRL 99, 235503 (2007), where a simple model (the Dzugutov potential) forms quasicrystals through self-assembly. In a sense even more extreme if they accept the range of hard interactions to be zero, is the amazing complexity of crystals formed by hard particles, again shown by Sharon Glotzer [Science 337, 453 (2012)], which can also form quasicrystals [Nature 462 773 (2009)]. Even binary spherical systems can form crystals with huge unit cells, such as the Frank-Kasper phase formed by the Wahnstrom binary Lennard-Jones model [PRL 104, 105701 (2010)].

So this idea that one can get away with small primitive cells is wrong in the end. The authors understand this, but their claim that one can get away with neglecting larger unit cells is already known to be incorrect in the colloid literature. So how can they claim that their method is somehow complete? It comes with lots of caveats - as do the existing methods. It's just that the

caveats are different.

In terms of making the manuscript readable, much remains to be done. At the moment it is much too long and too technical, yet simultaneously manages to leave out key results that show the potential importance of these results for the non-specialist, appropriate for a journal like Nature Communications. In particular, SI Fig. 7 should be in the main text. The simplicity of the interaction potentials with which the authors have obtained these results is significant. So show the interaction potentials in the main text!

I suggest the authors think hard about what they really want to say. Much of the material should take the form of a follow-up paper in a more specialized journal. Indeed, the current manuscript reads as if it is targeted at a more specialized audience. But the authors need to address the issue of 3d properly instead of the dismissive way they have done so. Of course 3d is more complex than 2d. But if this method is really to have any impact, it will be in 3d. I can scarcely think of any non-trivial colloid self-assembly paper in 2d experiment [with the exception of the work by Granick noted above], rendering the current 2d results as interesting, but more of a curiosity than the future of self-assembly that the authors claim. For a small enough primitive cell, there is no reason as far as I can see that 3d should be harder than 2d, and in any case, as noted above, the authors claims that small primitive cells are "OK" for short-ranged interactions are rather flimsy to put it mildly: in short they need to town down their claims so that they are correct and acknowledge that their method - in its practical implementation - has limitations and therefore (like the other methods) will only access a subset of possible structures. This is not what the seem to claim in the introduction at the moment.

Reviewer 1 (Remarks to the Author):

I have read the manuscript “Using symmetry to elucidate the importance of stoichiometry in colloidal crystal assembly” by Mahynski et al. with a great interest.

We thank the reviewer for their interest in our work and their thoughtful review.

The authors propose that the use of multicomponent systems with very simplified isotropic interparticle interactions can generate a number of complex 2D architectures. I, however, doubt how realistic are the assumptions used in the simulations. Excluded volume entropy seems to be completely neglected in the simulations. Entropy is the only driving force in crystallisation of hard-wall colloids and it is an essential driving force for the majority of colloidal systems exhibiting spontaneous crystallisation. Assuming purely enthalpy-driven situation severely limits the applicability of the model to experimentally-realizable colloidal systems.

Excluded volume is **not** neglected in our approach. It is included in several ways. First, our Lennard-Jones-esque potential diverges creating a barrier at roughly the particle diameter, so all potential energy calculations include this repulsion. Furthermore, in our structure generation routine, the spacing of the lattice points upon which colloids are placed is set such that points are spaced roughly equal to where particles contact their neighbor(s). This is further relaxed during the basin hopping stage to find the optimal spacing, which accounts for the specific form of the potential chosen in the model. We absolutely agree that excluded volume is an important part of crystallizing colloidal systems, and is indeed the only driving for purely repulsive hard-sphere systems; however, for hard-sphere systems with no energetic interactions, crystallization is driven by increasing the system pressure (decreasing volume). When colloids are driven to crystallize by grafting attractive ligands to the surface, high pressures/densities are not needed. This is how nearly all assembly occurs with DNA-grafted nanoparticles, etc. In these cases, assembly occurs at low densities and ambient temperature and pressure. The resulting lattice is generally the one which maximizes the contacts between particles to minimize the energy. This is what is meant by “enthalpy-driven” crystallization and has been validated for many atomic and molecular systems experimentally and theoretically; for example, we refer the reviewer to some of our included references: Macfarlane *et al. Science* (2011), Vo *et al. PNAS* (2015), Song *et al. Langmuir* (2018).

Do I understand it correct that Equation (5) is applicable also for $r < \sigma$? No hard wall interaction assumed?

See our response above. Yes this equation is valid for $r < \sigma$, and this potential **does** create essentially a hard wall at the particle diameter. We refer the reviewer to Fig. 7 in the previous version of the SI, which is now Fig. 3(a) in the main text of our revised manuscript. We have moved this figure to the main text to clarify this point for future readers as well.

The present simulation results are limited to 2D. Is there any experimentally-realizable

analogue that can test the current predictions?

In Song *et al. Langmuir* (2018) such an experimental system was created, and qualitatively validates our predictions. We chose to work with this particular system and model (“multiflavored” colloidal particles) because of our familiarity with the system, and previous validation. We can also refer the reviewer to additional theoretical and experimental papers on this system, which validate many of our predictions, though our simulations cover a much broader parameter space than what has been experimentally tested to date: Scarlett *et al. Soft Matter* (2011), Casey *et al. Nat. Comm.* (2011), and Mahynski *et al. Soft Matter* (2017).

As I am not performing simulation myself, it is difficult for me to judge the advantage of the computational approach such as the constraint satisfaction problem. The observations are interesting. However, it is not clear to what extent they could be experimentally realised as the interparticle interactions seem to be unrealistic for colloidal particles.

We refer the reviewer to our responses above for systems where the chosen model has proven to be at least qualitatively correct. Furthermore, we emphasize that our choice of a “Lennard-Jones-esque” potential form was made because it represents a simple, but flexible mathematical framework to work with, where a single parameter controls the relative attraction/repulsion between pairs of species. Our method of candidate generation, followed by refinement with basin-hopping does not depend on this choice of the potential. We would also like to point out that similar potentials have been used to successfully model other experimental DNA-based systems which are not multi-flavored. This citation [Auyeung *et al. Nature* (2014)] has been added to the text for the benefit of the reader as well. Our adoption of a simple form makes the results more generally applicable, illustrating the existence of a new qualitative effect (crystal design via phase separation driven by stoichiometric control) that was not previously realized, which is the main scientific conclusion of this work.

Which specific experimental systems this effect can manifest in and to what degree is the subject of future work, and does not detract from the exciting revelation that such control is even within the realm of possibility. The point is that even simple potentials can give rise to interesting lattices, so we believe this serves as evidence that more complex, realistic (albeit system-specific) ones may be capable of this also. Again, the main point of our paper is that this method allows us to generate an ensemble of structures and screen them to compute phase diagrams, enabling design of complex crystals containing many components with simple interactions instead of a single one with complex (potentially unrealizable) interactions.

In addition, spontaneous formation of complex structures by colloids with isotropic interactions was earlier demonstrated even in a single-component systems, using a combination of short-range attraction and long-range repulsion. I missed mentioning of these results in the introduction.

Indeed, so-called “SALR” potentials can also give rise to various interesting structures, further illustrating another specific class of potentials that can be investigated in the future, but whose focus would detract from the generality of our conclusions. We have added the following citations to these types of potentials: Ciach *et al.* *Soft Matter* (2013), Godfrin *et al.* *Soft Matter* (2014).

To summarize, I vote against publication of this MS in Nature Communications and suggest to send it to a journal specialised in computational techniques. At the same time, I encourage the authors to consider expanding their approach towards more realistic models, that could prove relevant for experimentally-realizable systems. Andrei Petukhov

In addition to the revisions undertaken as described above, we have further revised the text and flow of the manuscript to focus on less technical aspects of the approach taken, and instead emphasize the broader scientific impact of the work. Specifically, that our results demonstrate a fundamentally new pathway for crystal design using stoichiometric effects to drive phase separation into different lattices, which is why it is of general interest to a broad audience; it is more than just a computational paper and we believe the revisions better reflect this fact.

We have made numerous revisions, but below we highlight a few examples to this end and where they have been made.

Introduction: We demonstrate how enthalpically dominated colloidal systems with only two or three components, interacting with simple isotropic potentials, can give rise to a wide range of structures, and how selection between close-packed and open structures can be performed by changing composition alone. Furthermore, the generality of our method suggests this tactic is applicable to a range of experimentally realizable colloidal systems and can provide useful routes to complex structures for the design of advanced materials.

Section “Stoichiometric Control”: Phase separation can, therefore, be harnessed as a powerful mechanism for controlling self-assembly. A system with a fixed set of inter-particle potentials that assembles into one structure out of a solution initially prepared at one composition, can give rise to a completely different structure when the solution is prepared with a different ratio of the same components. In this way, a single system can be designed so that simply by varying the solution mixing ratio of constituents, a number of structures with different stoichiometries can be produced.

Conclusions: It is important to highlight the general applicability of both the presented method and the results regarding stoichiometric control. Although we have focused on presenting results from two-dimensional systems, the concepts presented here are extensible to higher dimensions as well. The interaction potentials considered here are very general, but experimental schemes for realization of such interactions in multi-component systems exist using multi-flavored DFPs. These DFP systems are not limited to two dimensions, and simulations and experiments in both two and three dimensions

have been performed on these systems to show the capacity of simple pairwise models to capture DFP assembly effects. They additionally demonstrate the feasibility of fine-tuning interactions in multicomponent mixtures as necessary to achieve self-assembly of particular structures. Finally, the results presented here have the potential to be particularly useful for physical realization of many superlattices including unique open structures, given that mixture stoichiometry is often easier to control than pairwise interactions, and has the potential to be just as powerful in terms of controlling structural ordering during self-assembly.

Reviewer 2 (Remarks to the Author):

This paper reports a new method for the search of two-dimensional colloidal crystals based on symmetry considerations. This is of course very interesting for the community of researchers working in crystal structure prediction. I leave to the editor to judge whether this paper can be of interest to a broader community. In any case, the paper is of high quality, so that I recommend publication after minor revision as specified below.

We thank the reviewer for their favorable assessment and agreement that these results will be of broad interest to the scientific community.

- Since this method is based on direct enumeration, I would like the authors to comment a little more deeply about its application to three-dimensional crystals, in which a third dimension may increase the number of possibilities in such a way that the method may become not applicable.

Three dimensional systems are of interest to us as well. We are optimistic that this will be applicable in 3D for several reasons, and it is the subject of ongoing work. While we do not wish to speculate in the manuscript, we will provide some discussion here. First of all, in 3D there are 230 space groups instead of 17, making the code significantly more involved. This is the reason we have not reported those results here, but plan to provide another contribution in the future. We have focused on 2D simply to show the reader a new pathway to designing colloidal crystals (via phase separation) is possible.

We believe 3D enumeration, which drives the technique, will be tractable to a reasonable extent. Our approach leverages the fact that lattice sites placed at the edges of the fundamental domain boundaries have fractional “contributions” to the domain, which provide the basis for formulating the constraint satisfaction problem. As discussed in the main text, the surface-area-to-volume (in 2D, perimeter-to-surface-area) ratio is essentially what determines how heavily these constraints reduce the number of possible configurations. As an example, consider the following comparison between an orthorhombic domain in 2 and 3 dimensions.

Assuming the “sides” of a domain are discretized into L sites along each axis, the ratio of the number of “surface” or “boundary” sites to those enclosed in the interior of the domain is:

$$r_2 = \frac{4(L - 1)}{(L - 2)^2}. \quad (1)$$

Similarly, in 3D:

$$r_3 = \frac{6(L - 1)^2}{(L - 2)^3}. \quad (2)$$

Considering the ratio of these two we have:

$$\frac{r_2}{r_3} = \frac{4L - 2}{6L - 1} < 1. \quad (3)$$

Clearly, this ratio is always less than 1, implying that in 3D a relatively higher fraction of the sites will fall at the boundaries than in 2D. Remarkably, this would seem to suggest that our technique might work even better in 3D than in 2D. However, obviously there will be more total sites in the 3D case which might offset this and make the overall number of solutions grow larger; there are also additional details involving the shape of the fundamental domains for each group which we have neglected here, but this simple example is illustrative. The net result of these two competing aspects of the problem remains to be seen, and as is the subject of future work. Regardless, the approach should still be effective up to some primitive cell size; the actual size itself will determine how helpful this approach is in relation to other methods.

- The claim that short-range interactions mean small unit cells is not really granted. I do not see what long-range interactions stabilize quasi-crystals for example.

We agree that the two do not strictly guarantee each another, and we have only argued from a rough scaling perspective, instead of one of more rigor, simply to provide justification for the fact that many crystals (especially those found in nature and most experimental colloidal systems) do, in fact, have relatively small unit (and also primitive) cells. There are some notable exceptions, however.

Indeed, it is still an outstanding question as to what characteristics of a potential lead to quasi-crystal (QC) formation. Relatively short ranged interactions can give rise to QCs. These QCs lack sufficient translational order to belong to any space/wallpaper group and as a result, are not present in the direct enumeration stage of our method. Of course, a library of known QCs can always be added to our set of generated structures during the screening stage, which is practically how we would envision incorporating these candidates into our method at that juncture. QCs represent a relatively small set of possible configurations; in the absence of strong alternatives, we feel our method still represents an important and powerful way to explore (most of) configurational space with relative ease.

It is also very important to point out that our approach only **begins** with these directly enumerated candidates. Subsequent basin hopping (stochastic) is **always** performed and this helps these crystalline “starting points” refine into their most stable ground state; recall that these are sometimes dispersed gases, clusters, etc. with lower symmetry than the starting point which can include QCs which form by relaxing from a crystalline parent structure. In practice, this should allow a system to find any QC or QC-approximants (which still have some translational order, i.e. p1, at some length scale) which might manifest in the ground state for any set of interactions. Thus, it is **possible** to discover QCs with our approach. In fact, the basin-hopping stage did

locate certain ground state structures with 5-fold symmetries that could be construed as a QC-approximant for several stoichiometries in the ternary mixture reported in the main text; however, in no case did those structures belong to the convex hull so are not thermodynamically stable. No QCs were found to be stable for any system we investigated with our very simple potential reported, but it is important to emphasize that our method is still capable of finding these candidates.

Furthermore, even if these candidates are missed altogether, our method is purely a predictive one. “Forward design” tests with molecular dynamics were always performed to validate these predictions and we did not find any evidence to invalidate our predictions reported in this work. While QCs are an important class of materials that has garnered renewed interest, we are mainly focused on studying crystalline systems, as reflected in the title of this manuscript. For this reason, the title was chosen carefully to directly state our approach is aimed at systems which are truly crystalline. We do not wish to confuse the reader into believing otherwise, so we have made the following changes to our manuscript to convey this point more clearly:

Section “Building Phase Diagrams”: **In fact, all stable periodic lattices reported in this work were found by direct enumeration, ultimately requiring no stochastic relaxation, demonstrating the robustness of this enumeration scheme.**

...

Note that lower structures which do not belong to any wallpaper group, such as quasicrystals or disconnected cluster phases, are not generally proposed in the initial candidate pool. A relaxation stage with basin hopping is important since it allows lower symmetry structures to emerge from higher symmetry parent structures.

Reviewer 3 (Remarks to the Author):

Report for Using symmetry to elucidate the importance of stoichiometry in colloidal crystal assembly by Mahynski et al.

The authors address the problem of elucidating possible crystal structures formed by multicomponent colloidal systems, in which the interactions can be tuned, and thus in principle the range of crystal structures accessible is very large. The authors claim to massively simplify this problem by considering the symmetries of all wallpaper groups in 2d, thus building a comprehensive set of structures. Notably, the use of symmetry as employed by the authors allows a spectacular drop in the number of structures that need to be considered. So if the results really are as impressive as the data seem to indicate, given the broad interest of colloidal (and nanoparticle) self-assembly, I think there is potential for this paper in a wide-ranging journal like Nature Communications.

We thank the reviewer for their favorable assessment of our work and its impact.

However, in its present form, the manuscript is not easy to understand and much remains to be done in order to render it in a suitably palatable form. To give just one example, I had to go carefully through the SI to obtain absolutely essential results to understand the main message of the manuscript. The authors have been over-ambitious in terms of the amount of material they have chosen to attempt to present. They absolutely must take a lot of material out (like 50% say) if they are to have any hope of reaching the kind of general audience that Nature Communications aims at.

The reviewer's point is well taken. To this end we have made a series of significant revisions to the manuscript which we will briefly summarize here. The main message of our manuscript is that stoichiometry can be used to fundamentally control the resulting crystal structure of a multicomponent mixture of colloids, and is as robust as tuning the interparticle potential(s) themselves under many instances. To show this we must first introduce our methodology (enumeration), then discuss how phase diagrams are computed, and finally, show representative results proving this claim. In the previous version of our manuscript, the first two sections were weighted too heavily, distracting the reader from the final conclusions. We have made significant revisions to the first half of the manuscript, in addition to those suggested below and above (by other reviewers). In summary, we have removed roughly 5 paragraphs worth of text and abbreviated figures 1 and 2, shortening the current manuscript by roughly 2 pages. A complete set of revisions is available in the included "diff.pdf" file. We believe these revisions make the main message of the paper more clear, and guide the reader more easily from the problem statement to the final conclusions we present.

(1) In their literature review the authors have missed major contributors to this problem - Peter Harrowell in computer simulation and theory - and Steve Granick in experiment. Granick's kagome lattices [Nature 469 381 (2011)] are among rare (i.e. unique as far as I know) as non-trivial 2d colloidal self-assembly.

We thank the reviewer for pointing out the work by Harrowell. We are familiar with the work by Granick, but in the interest of brevity (since this is not experimental work), had omitted that citation. We have now added both of these citations for the benefit of the reader.

Meanwhile Harrowell has a series of papers with Toby Hudson, which tackled the same problem in 3d, starting from the set of all known crystals that form in atomic systems, focusing on inorganic materials, such as alloys. While of course such an approach is limited in terms of the dataset, one may observe that nature has had some very considerable time to sample the phase space. Examples include J. Phys. Chem. B 112, 10773 (2008).

Harrowell has explored other ways to reduce the configurational space than the approach of the authors - with Pierre Ronceray, he has used a lattice model to elucidate entire sets of structures in 3d: arXiv:1606.02579

Harrowells methods are important, not least because they pertain to 3d. I accept that 2d is much easier for the authors to work with than 3d, but they cannot be as dismissive of 3d as they are in the current version of the manuscript.

Indeed, these methods by Harrowell and Hudson are certainly relevant for 3D. We have added these references for the benefit of the reader. Since the focus of this paper is 2D we did not initially include extensive references to 3D work; however, we did not intend to be dismissive of the importance of 3D systems, only to be direct with the reader about the scope of this work. Future extension of this work into 3D is planned, but beyond the scope of our current objective. A more detailed comment on the application of our method to 3D systems may be found in the comments to Reviewer #2, who also raised this point.

Surely for sufficiently small primitive cells, 3d is tractable. As for the relevance of the size of primitive cell for sufficient completeness, please see below - the authors claim that the size of the cells they consider is sufficient is far from justified. The authors should at least demonstrate that their approach is pertinent to 3d, by taking some small primitive cells.

A similar question was raised by Reviewer #2; our response is given above. We reiterate that in 3D there are 230 space groups not 17, each of which contains fundamentally more information at an algorithmic level than wallpaper groups in 2D do. This precludes an immediate implementation of a 3D version of our method - it is more than just the size of the primitive cell, though as we discussed in our response to Reviewer #2, there is good reason to anticipate feasibility in 3D. Such work is planned, however, and there is nothing in principle that prevents the concepts we have presented here from being applied; albeit with a significantly larger coding effort. We reiterate that the scientific conclusion we are seeking to present here is that this enumeration/relaxation procedure enables the investigation of the phase behavior of multicomponent systems; it is not contingent on the dimensionality of the system being studied. This investiga-

tion has shown that this method provides a novel route to controlling the structure of self-assembling colloidal systems that was previously thought only to be a minor factor in tuning the resulting structure. As this reviewer points out later, the previous length of the manuscript obscured this point from the reader and we hope that our substantial revisions have made this point more clear now.

(2) A major concern with the manuscript is the size of the primitive cell. The authors claim this should be somehow connected to the interaction range. This cannot be correct. See for example PRL 99, 235503 (2007), where a simple model (the Dzugutov potential) forms quasicrystals through self-assembly. In a sense even more extreme if the accept the range of hard interactions to be zero, is the amazing complexity of crystals formed by hard particles, again shown by Sharon Glotzer [Science 337, 453 (2012)], which can also form quasicrystals [Nature 462 773 (2009)]. Even binary spherical systems can form crystals with huge unit cells, such as the Frank-Kasper phase formed by the Wahnstrom binary Lennard-Jones model [PRL 104, 105701 (2010)].

So this idea that one can get way with small primitive cells is wrong in the end. The authors understand this, but their claim that one can get away with neglecting larger unit cells is already known to be incorrect in the colloid literature. So how can they claim that their method is somehow complete? It comes with lots of caveats - as do the existing methods. Its just that the caveats are different.

We agree with the reviewer. Our previous argument that the primitive cell size and interaction range are equivalent somehow is sometimes, but not universally, justified. We did not intend to present this as a rigorous statement, but only to argue the point that it is reasonable to consider small to moderately sized primitive cells in many cases. Note that for an interaction potential with a cutoff of 3, then up to a 30x30 cell would be considered within the bounds of our scaling argument we previously presented; this would include those systems referenced above. A cell this large does go beyond the reasonable level of enumeration with this method; however, the scaling argument was made only to emphasize that there exists a non-trivial limit (one which extends over a reasonable range), up to those sometimes used to study fluids in molecular simulation, *i.e.*, 8x8 cell, in which exhaustive enumeration is still possible with our approach, but impossible with random structure searching. Because this was confusing, we have simply removed the scaling argument altogether to prevent the reader from being misled. Even if the method cannot enumerate literally **all** possible cells, a large, reasonable set is expected to provide reasonable results on which to base the phase diagram calculation (which would otherwise not be feasible anyway). This provides the basis for deeper investigation and testing with other techniques, such as molecular dynamics, which we did and have presented in both the main text and in the SI.

In terms of making the manuscript readable, much remains to be done. At the moment it is much too long and too technical, yet simultaneously manages to leave out key results that show the potential importance of these results for the non-specialist, appropriate for a journal like Nature Communications. In particular, SI Fig. 7 should

be in the main text. The simplicity of the interaction potentials with which the authors have obtained these results is significant. So show the interaction potentials in the main text!

We agree and thank the reviewer for this perspective. We have moved this to the main text as Fig. 3a. To our estimation, our manuscript previously fell within the length guidelines, although the content was not sufficiently streamlined for a general audience. Following this reviewer's advice we have undertaken a significant revision of Figures 1 and 2, and have revised the text to remove roughly 2 pages of text. As discussed in our response above, we believe this significantly improves the flow of the text from problem statement, to a discussion our (new) method, to results which illustrate a new paradigm for controlling colloidal crystal assembly. This is the main point of the paper and we have endeavored to make this point more salient than the method by which it is arrived at.

I suggest the authors think hard about what they really want to say. Much of the material should take the form of a follow-up paper in a more specialized journal. Indeed, the current manuscript reads as if it is targeted at a more specialized audience. But the authors need to address the issue of 3d properly instead of the dismissive way they have done so. Of course 3d is more complex than 2d. But if this method is really to have any impact, it will be in 3d.

While 3D systems are certainly an important area of research, which we do not mean to be dismissive of, there are many applications for 2D materials. We failed to previously highlight these areas in which we anticipate our approach will have an impact, so we have added the following to the introduction to clarify this point:

Furthermore, there are many important technological applications for ordered two-dimensional materials including interfacial films, monolayers, porous mass separating agents, and structured substrates which require careful tuning of their crystalline structure.^{44–47} Epitaxial growth and layer-by-layer assembly also require a detailed understanding of two-dimensional precursors to grow three-dimensional crystals.^{48–50}

I can scarcely think of any non-trivial colloid self-assembly paper in 2d experiment [with the exception of the work by Granick noted above], rendering the current 2d results as interesting, but more of a curiosity than the future of self-assembly that the authors claim. For a small enough primitive cell, there is no reason as far as I can see that 3d should be harder than 2d, and in any case, as noted above, the authors claims that small primitive cells are OK for short-ranged interactions are rather flimsy to put it mildly: in short they need to tone down their claims so that they are correct and acknowledge that their method - in its practical implementation - has limitations and therefore (like the other methods) will only access a subset of possible structures. This is not what they seem to claim in the introduction at the moment.

We agree, and did not mean to convey this (see response to reviewer 2), we have made the following changes, among many:

- The “completeness of coverage” we refer to in the manuscript is the that proposed candidates cover all possible space groups, not that it is capable of enumerating all local minima (structures) in the free energy landscape; enforcing full coverage of all symmetries is something that has been a challenge to achieve in other inorganic crystal structure prediction problems, such as with metal organic frameworks, and our method is not subject to such concerns. This is reiterated in the conclusions, and elsewhere: **This approach serves as an efficient way to explore all possible wallpaper groups and ensure that the ground state is discovered.**
- In “Building Phase Diagrams”: **Note that lower symmetry structures which do not belong to any wallpaper group, such as quasicrystals or cluster phases, are not generally proposed in the initial candidate pool. A relaxation stage with basin hopping is therefore important since it allows these lower symmetry structures to emerge from higher symmetry parent structures. ... In fact, all stable periodic lattices reported in this work were found by direct enumeration, ultimately requiring no stochastic relaxation, demonstrating the robustness of this enumeration scheme.**

The introduction has been significantly revised to make the main scientific message of our paper more clear. Namely, that stoichiometric control over phase separation represents a novel route to controlling the assembly of complex, multicomponent mixtures which has not previously been viewed as a powerful/useful approach. Any caveats in the method by which we explore this do not detract from this final result as all predictions made were assessed by “forward design” molecular dynamics simulations. Both the order and content of the paper have been revised to make this more clear to the reader, and in the process, also more transparent about the scope and aims of the work.

REVIEWERS' COMMENTS:

Reviewer #1 (Remarks to the Author):

I would like to thank the authors for serious consideration of the previous comments of the referees. I feel that many important points are better clarified now and the readability of the paper is improved for broader readership of Nature Communications. I still feel that the entropic contributions must be better accounted for in forthcoming publications. For the time being I am sufficiently convinced by the authors. The observed rich library of superstructures observed in a relatively simple model is an interesting result, which can stimulate further theoretical and experimental studies. I am therefore pleased to recommend the MS for publication in Nature Communications. Andrei Petukhov

Reviewer #2 (Remarks to the Author):

The reply is convincing. I recommend publication.

Reviewer #3 (Remarks to the Author):

Second report for "Using symmetry to elucidate the importance of stoichiometry in colloidal crystal assembly" by Mahynski et al.

The authors have done a reasonable job of addressing the points raised in the first round of reviewing, and indeed the manuscript is much shorter and more accessible. I am now happy to recommend the manuscript for publication.